# Partial or complete blocking of gD binding to HVEM affects primary and latent HSV-1 infection as well as eye disease in HSV-1 infected mice

Deepak Arya, Ujjaldeep Jaggi, Jay J. Oh, Shaohui Wang, Homayon Ghiasi ⓘ *

Department of Surgery, Center for Neurobiology and Vaccine Development, Ophthalmology Research, Cedars-Sinai Medical Center, Cedars-Sinai Health Sciences University, Los Angeles, California, United States of America

* homayon.ghiasi@csmc.edu, ghiasih@cshs.org

## Abstract

Herpes simplex virus type 1 (HSV-1) glycoprotein D (gD) engages herpesvirus entry mediator (HVEM) and other cellular receptors to mediate efficient viral attachment. We recently identified gD amino acids (aa) Q27-T29 as critical to completely block gD binding to HVEM, while the single Q27P mutation in gD reduced its binding to HVEM by approximately 80%. Thus, we constructed two mutant viruses: one with a single gD aa mutation (v27) and the other with three gD aa mutations (v27-29) in the HSV-1 strain GFP-McKrae background. These mutant viruses clearly demonstrated the consequences of disrupting gD binding to HVEM, which reduced HVEM expression and altered the expression of other receptors. To determine whether these mutations also affected viral replication, ocular disease severity, latency levels, and reactivation kinetics, mice were ocularly infected with WT, v27, or v27-29 viruses. Both mutant viruses exhibited significantly reduced viral replication in tears compared with the WT virus, even after corneal scarification. The v27-29 virus showed a greater reduction in viral replication than the v27 virus. Mice infected with both mutant viruses had significantly lower levels of eye disease, latency, and T cell exhaustion than in WT virus-infected mice. In the absence of scarification, significantly fewer trigeminal ganglia (TG) were reactivated by mutant viruses than by the WT virus. In contrast, after scarification, all TG from WT-, v27-, and v27-29-infected mice were reactivated, and the time to reactivate was similar between the two mutant viruses, but reactivation was significantly less delayed than in WT-infected mice. Together, these findings demonstrate that disrupting gD-HVEM binding reduces viral virulence and limits latency, reactivation, and T cell exhaustion. Thus, the Q27-T29 region of gD is a critical determinant of infection dynamics and a potential therapeutic or vaccine target.

**Data availability statement:** All relevant raw data are in the manuscript.

**Funding:** This work was supported by Public Health Service grants RO1EY029160, RO1EY26944, and RO1EY033574 from the National Eye Institute to HG. The funders had no role in study design, data collection and analysis, decision to publish, or preparation of the manuscript.

**Competing interests:** The authors have declared that no competing interests exist.

## Author summary

We previously showed that the absence of HVEM significantly affects reactivation of HSV-1 from latency but not primary ocular infection. We also found that complete blockage of gD binding to HVEM requires mutations of amino acids 27–29, while a single Q27P mutation reduces binding by approximately 80%. Therefore, this study focused on defining mechanisms linking partial and complete gD binding to HVEM to ocular viral replication, ocular disease, latency establishment, subsequent reactivation in vivo, and T cell exhaustion in TG of latently infected mice. Partial gD binding to HVEM, with and without corneal scarification results in less viral replication in the eye, latency, and T cell exhaustion than WT virus carrying full-length gD. A large number of TG from latently infected mice without scarification did not reactivate, and all TG from mice infected with the two gD mutant viruses reactivated after scarification. Thus, complete blocking of gD binding to HVEM provides a framework for developing strategies to prevent or significantly reduce virus replication in the eye and consequently, reactivation, providing a new tool to minimize recurrent eye disease.

## Introduction

Herpes simplex virus type 1 (HSV-1) is a neurotropic, double-stranded DNA virus with broad tropism due to its use of diverse host receptors including herpesvirus entry mediator (HVEM, *TNFRSF14*), Nectin-1, Nectin-2, 3-O-sulfated heparan sulfate, paired immunoglobulin-like type 2 receptor (PILRα), non-muscle myosin heavy chain IIA (NMHC-IIA), and myelin-associated glycoprotein on mucocutaneous, ocular, and labial tissues [1–5]. This receptor versatility renders HSV-1 a leading cause of infectious corneal blindness worldwide [6,7]. HSV-1 glycoproteins bind to these receptors, thereby enabling entry and replication in infected cells. Thus, the coordinated engagement of multiple viral glycoproteins and distinct host receptors determines the scope of eye disease and latency-reactivation. gD is one of more than 80 HSV-1 genes, is an essential virus glycoprotein, and the first gene that was used for vaccine studies [8–16]. It is well established that it plays a key role in virus infection and pathogenesis and is a major inducer and target of humoral and cell-mediated immune responses following infection [17–19].

HVEM, a key gD-binding partner, is expressed on corneal epithelial cells, the nervous system, lungs, and intestines, as well as on immune cells, including T cells, B cells, natural killer cells, dendritic cells, and myeloid cells [20]. As a TNFR superfamily member with immunomodulatory functions, HVEM signals through ligands such as BTLA, LIGHT, CD160, and lymphotoxin α (LTα) to shape local immune responses, serving as a molecular switch between proinflammatory and inhibitory signaling that aids in establishing homeostasis [21,22]. Selective upregulation of HVEM, but not other HSV-1 entry receptors, in latently-infected TG highlights HVEM's importance in latency and reactivation, as shown by *HVEM*[-/-] mice [23]. In contrast to *HVEM*[-/-] mice, *BTLA*[-/-]*, LIGHT*[-/-]*,* or *CD160*[-/-] mice showed reduced latency but enhanced reactivation

associated with increased HVEM expression, suggesting that disruption of HVEM-ligand homeostasis alters the balance between latent infection and reactivation in a gD-dependent manner [24].

Previously, using WT and mutant gD-expressing viruses, we showed that gD-HVEM interaction plays a significant role in HSV-1 infectivity. In contrast to WT McKrae, which does not need corneal scarification for efficient replication in the eye of infected mice, parental viruses- KOS, for these gD mutants, required corneal scarification due to their lower infectivity *in vivo* [25]. Thus, we generated gD mutant viruses in the GFP-McKrae background. Previously, it was shown that gD binds to HVEM via gD amino acid (aa) 27 by immunoprecipitation (IP) and ELISA [26,27]. Thus, to confirm the role of gD aa 27 binding to HVEM, we mutated aa 27 (Q27P) and tested its lack of binding to HVEM using a combined IP and western blot pull-down as we described previously for gK and UL20 [28,29]. Notably, mutations of gD aa 27–29 directly target a key contact zone where this segment pairs with HVEM aa 35–37 to form a β-sheet interface, whose stability relies on hydrogen bonds, van der Waals contacts, and electrostatic interactions. In our previous report, pull-down assays showed significantly less gD(Q27P) binding to HVEM than to the WT gD control, while mutation of gD aa 27–29 completely blocked gD binding to HVEM [30]. These results suggest that more than one aa is involved in the functional binding of gD to HVEM. We therefore constructed two gD mutant viruses containing either a single aa mutation (v27) or three aa mutations (v27-29) in the HSV-1 strain GFP-McKrae and characterized them *in vitro*. Replication of both mutant viruses was similar to that of the WT virus in multiple mammalian cell lines, but not in Neuro2a cells.

In this study, we examined the functional impact of partial and complete inhibition of gD binding to HVEM by using these gD-mutant viruses in an *in vivo* model of ocular HSV-1 infection. Mice were infected ocularly, in the presence or absence of corneal scarification, with WT, v27, or v27-29 viruses. Viral replication, corneal disease, latency, T cell exhaustion, and reactivation were measured. Mice infected with v27 or v27-29 showed significantly less replication than mice infected with WT virus, while replication of v27-29 virus was significantly less than either v27 or WT viruses, suggesting that virus replication is directly proportional to gD-HVEM binding. Significantly less corneal scarring and angiogenesis were seen in v27 and v27-29 infected mice than in WT-infected mice, with or without corneal scarification, and scarification did not enhance eye disease. Further, significantly less expression of LAT, CD8α, and PD-1 was seen in TG from mice latently infected with v27 and v27-29 viruses than with WT virus. In non-scarified mice, a significant number of TG from mutant viruses did not reactivate. Whereas, in scarified mice infected with WT virus, all TG reactivated, but reactivation was delayed in v27, or v27-29 infected mice. Collectively, these data suggest that blocking gD-HVEM binding directly affects the HSV-1 life-cycle, with gD serving as a direct regulator of HVEM that can be manipulated to control HSV-1.

## Results

### Virus replication in tears from WT, v27, and v27-29-infected mice

Different HSV-1 strains, such as KOS, F, McKrae, 17, H129, SC, and ANG that are routinely used *in vivo*, have significant pathogenic differences [31]. We routinely use the McKrae strain, isolated from a patient with herpes simplex keratitis, because it exhibits a higher frequency of spontaneous and induced reactivation. Previously, we showed that HVEM-binding-deficient mutants, such as KOS-Rid1 and KOS-Rid2 regulate the latency-reactivation cycle by impairing gD–host receptor engagement [25]. These mutant viruses have been used to study gD binding to HVEM [26,27]. In contrast to McKrae, which replicates efficiently in the eyes of infected mice and does not require corneal scarification, strain KOS, originally derived from a patient with a cold sore, requires corneal scarification for efficient replication. However, the requirement for corneal scarification in these viruses limits their relevance to natural infection. To overcome issues associated with corneal scarification, we constructed two mutant viruses in the GFP-McKrae background, one containing a single aa mutation in gD (v27) and a second virus with 3 aa mutations in gD (v27-29). We fully sequenced v27 and v27-29 to confirm: 1) their similarity to each other and to the parental virus except for the noted aa mutations; and 2) that they replicate similar to WT virus *in vitro*, except in Neuro2a cells, which have neuronal origin [30].

To test the effect of partial and complete blocking of gD binding to HVEM *in vivo*, we ocularly infected mice with $2 \times 10^5$ pfu/eye of WT, v27, or v27-29 virus without corneal scarification. Tear films from 20 eyes per group in two separate experiments were collected on days 1–5 post-infection (PI) and the amount of virus was determined by standard plaque assays on Rabbit skin (RS) cells (Fig 1A). Virus replication in eyes of v27 and v27-29 infected mice was similar and replication in both was significantly lower on days 2, 3, and 4 PI than in WT-infected mice (Fig 1A, $P < 0.006$). Because the mutant viruses had significantly less viral replication than WT, another group of mice was similarly infected ocularly with the three viruses after corneal scarification (Fig 1B). Viral titers in the eyes of infected mice were determined as above. After

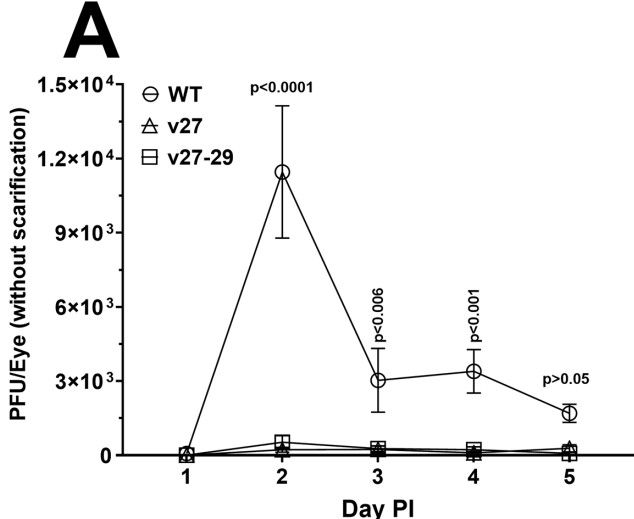

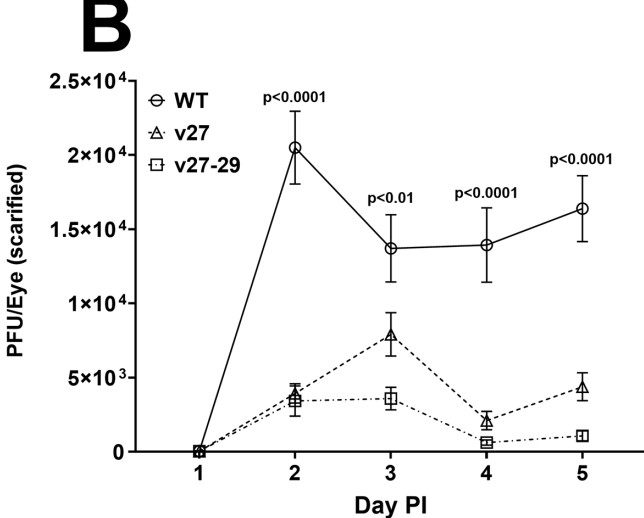

**Fig 1. Virus titers in the eye during primary infection.** Mice were infected ocularly with $2 \times 10^5$ pfu/eye of WT, v27, or v27-29 virus without (panel A) or with (panel B) scarification. WT, v27, and v27-29 viruses represent GFP-McKrae virus with wild-type, Q27P, or Q27A-L28A-T29A substitution of gD, respectively. Tear films were collected on days 1-5, and virus titers were determined by standard plaque assay. Each point represents the mean titers from 20 eyes ± SEM from mice with no scarification (panel A) or 24 eyes ± SEM from mice with corneal scarification (panel **B**). No scarification experiments were repeated twice.

scarification, virus titers in the eyes of v27 and v27-29 infected mice were higher (Fig 1B) than those of non-scarified mice (Fig 1A), but even after scarification, v27 and v27-29 infected mice had significantly lower titers on days 2–5 than did mice infected with WT virus (Fig 1B, P < 0.0001). Interestingly, on day 3 PI, v27-29-infected mice had significantly lower titers than v27-infected mice, highlighting differences in their viral growth (Fig 1B, P < 0.024). Thus, both partial and complete blockage of gD-HVEM interaction significantly reduced viral replication in the eyes of infected mice, even after corneal scarification, with the effect being more pronounced with v27-29 than with v27. Consistent with our previous study, in which replication of KOS-Rid1 and KOS-Rid2 mutants was modestly lower than the parental KOS virus (although the difference was not significant), the current study reproduced this trend, revealing an even more pronounced phenotypic difference.

**Effect of partial or complete blocking of gD-HVEM interactions on survival and eye disease in infected mice**

A total of 30 mice from two separate experiments were infected with 2 x $10^5$ pfu/eye of v27, v27-29, or WT virus, without corneal scarification. By day 28 PI, 29 of 30 WT and v27 infected mice survived ocular infection, while 30 of 30 mice infected with v27-29 survived infection. Corneal scarring (CS) and angiogenesis in surviving mice were measured on day 28 PI. Significantly less CS was observed in mice infected with v27 and v27-29 viruses than in mice infected with WT virus (Fig 2A, p < 0.0001), while no significant differences in CS were seen in v27 and v27-29 infected mice despite lower CS in v27-29 infected mice (Fig 2A, p > 0.05). We also assessed angiogenesis in eyes of the above-infected mice on day 28 PI. As in CS, significantly less angiogenesis was seen in mice infected with v27 and v27-29 viruses than in mice infected with the WT virus (Fig 2B; p < 0.0005). Although angiogenesis was lower in v27-29 infected mice than in v27 infected mice, these differences were not statistically significant (Fig 2B, p > 0.05).

We examined the effects of corneal scarification in groups of 12 mice infected as above, with 2 X $10^5$ pfu/eye of v27, v27-29, or WT virus. 9 of 12, 12 of 12, and 10 of 12 mice infected with v27, v27-29, or WT virus, respectively, survived ocular infection at day 28 PI. CS and angiogenesis in surviving mice were measured as above. Both CS (Fig 2C, p < 0.0001) and angiogenesis (Fig 2D, p < 0.0001) scores were significantly lower in v27 and v27-29 infected mice than in mice infected with WT virus. Levels of CS (Fig 2C, p > 0.05) and angiogenesis (Fig 2D, p > 0.05) were similar in v27 and v27-29 infected mice, with significantly more CS and angiogenesis after corneal scarification (compare Fig 2A and 2B with Fig 2C and 2D). These data suggest that blocking the interaction of gD with HVEM influences the course of eye disease in infected mice.

**Effect of blocking gD binding to HVEM on HSV-1 latency in infected mice**

To determine how partial or complete blocking of gD binding to HVEM affects latency, mice were infected with 2 X $10^5$ pfu/eye of v27, v27-29, or WT viruses without corneal scarification (Fig 3A). Individual TG was isolated on day 28 PI, total RNA was isolated, and TaqMan qRT-PCR was used to quantify viral LAT RNA. Mice infected with v27 or v27-29 viruses had significantly less LAT RNA than the WT virus.(Fig 3A, p < 0.05). Latency in v27-29-infected mice was lower than in v27 infected mice, but this difference was not statistically significant (Fig 3A; p > 0.05, v27 vs v27-29). We previously showed that the Rid1 mutant induced significantly less latency than the parental KOS strain [25], and our current data reproduced this trend, with a more pronounced difference.

To determine how corneal scarification affects latency in the absence, or reduced binding, of gD to HVEM, mice were infected after corneal scarification, as above with v27, v27-29, or WT viruses (Fig 3B). LAT expression in latently-infected mice was determined as above. LAT expression in infected mice after corneal scarification was approximately twofold higher (Fig 3B) than in non-scarified mice (Fig 3A). Similar to non-scarified mice, mice infected with v27 or v27-29 viruses had significantly less LAT RNA than mice infected with WT virus (Fig 3B; p < 0.0004). Latency levels in v27 and v27-2 infected mice were similar and not statistically significant (Fig 3B; p > 0.05, v27 vs v27-29). These results suggest that partial or complete binding of gD to HVEM significantly reduces latency in infected mice, irrespective of scarification.

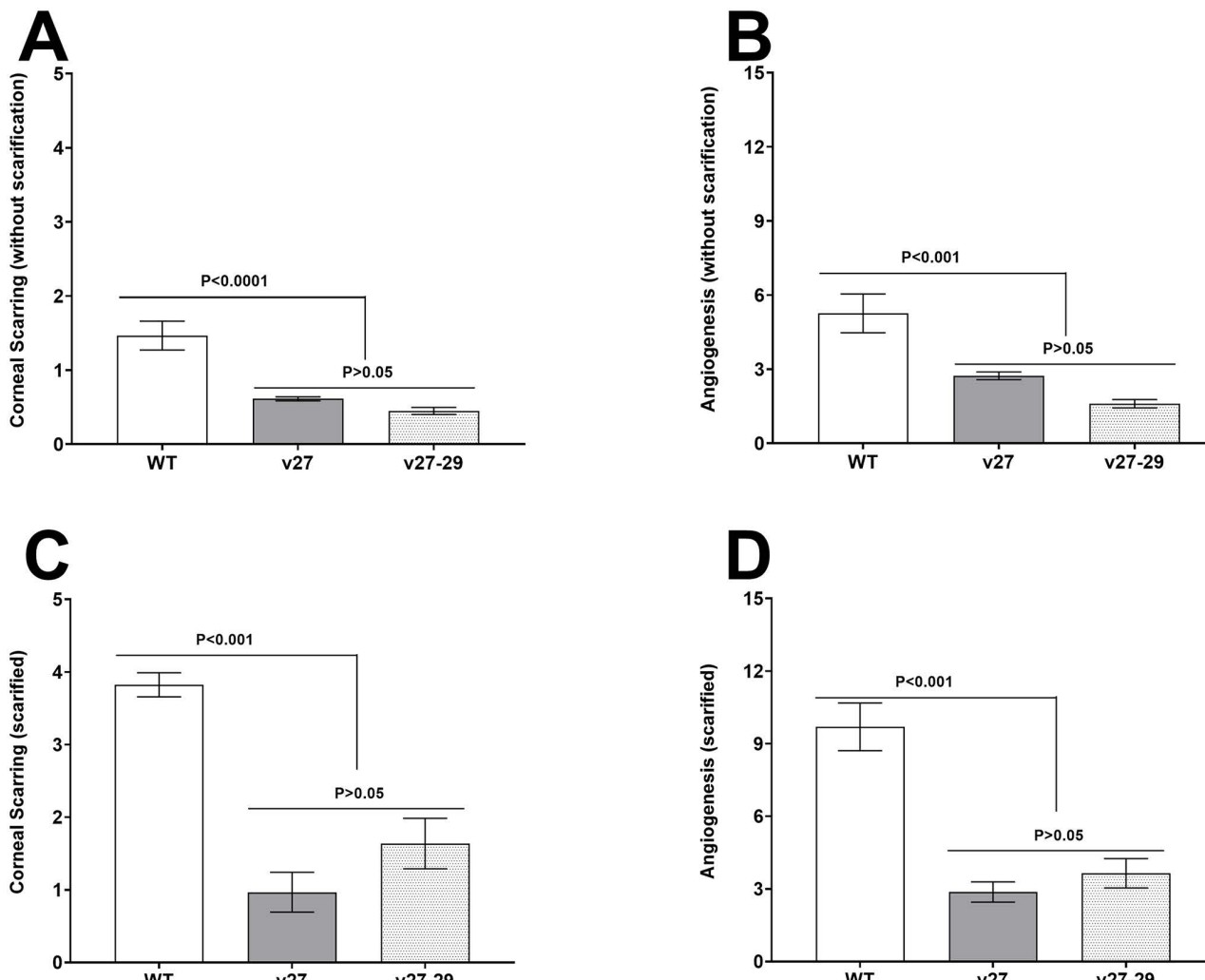

**Fig 2. Loss of gD binding to HVEM affects corneal scarring (CS) and angiogenesis in ocularly infected mice. (A-B)** CS and angiogenesis in mice without scarification. CS and angiogenesis in infected mice were measured on day 28 PI from WT-, v27-, or v27-29-infected mice. Each point represents the mean score±SEM from 58 eyes. Experiments were repeated 3 times. **(C-D)** CS and angiogenesis in surviving mice with scarification. CS and angiogenesis in infected mice were measured on day 28 PI. Data are from 17, 23, and 20 eyes from WT-, v27-, and v27-29-infected mice, respectively. Each point represents the mean score±SEM.

### Detecting gD and GFP transcripts in TG of latently-infected mice

Studies in mice and humans have shown that viral transcripts are expressed at very low levels in latently-infected TG [32–37]. All three viruses used in this study express GFP [30,38]; thus, we asked whether gD binding to HVEM affects GFP or gD expression in TG of latently-infected mice. To examine GFP and gD expressions in TG during latency, RNA isolated from TG of infected mice described above was analyzed by qRT-PCR and copy numbers were normalized to the endogenous GAPDH control. Using RNA from TG of mice infected with the three viruses in the absence of scarification, qRT-PCR analysis showed less GFP (Fig 4A) and gD (Fig 4B) expression in TG from v27 and v27-29-infected mice than from WT-infected mice, with no significant GFP or gD expression among the viruses (Fig 4A and 4B, p>0.05). Results were similar for GFP (Fig 4C) and gD (Fig 4D) expression in TG of WT-infected mice being higher than in v27 and v27-29

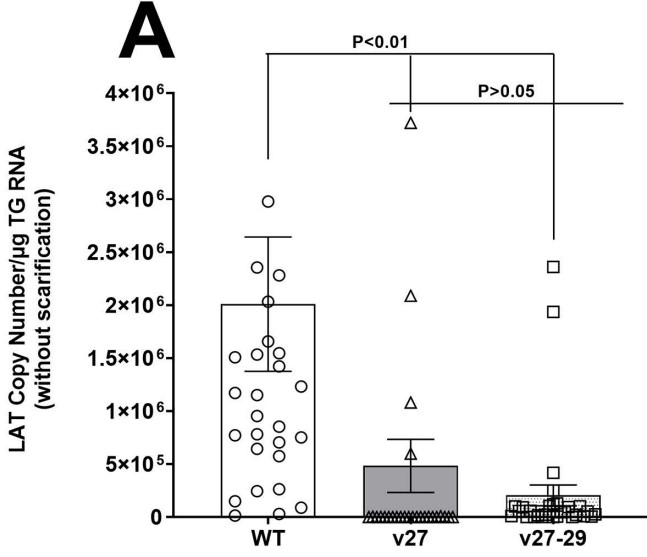

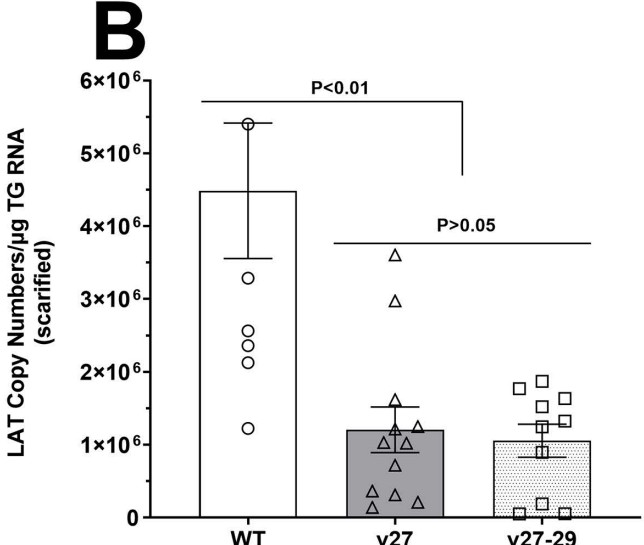

**Fig 3. Disrupting gD binding to HVEM affects LAT expression in TG of latently-infected mice. (A)** LAT expression in TG of latently-infected mice without scarification. Mice were ocularly infected with 2 X 10⁵ pfu/eye of WT, v27, or v27-29 virus. On day 28 PI, individual TG was harvested, and qRT-PCR was performed on total RNA from each TG. Estimated relative LAT copy number was calculated using standard curves generated from pGEM-5317. GAPDH expression was used to normalize LAT RNA expression. Data are from 29, 28, and 30 TG from WT-, v27-, and v27-29-infected mice, respectively ± SEM. **(B)** LAT levels following corneal scarification in infected mice. Mice were infected as above, but after corneal scarification. LAT copy number was calculated as above. Data are from 9, 12, and 10 TG from WT-, v27-, and v27-29-infected mice, respectively, ± SEM.

infected mice after corneal scarification. Thus, consistent with our previous studies [23,24], the absence of gD interaction with HVEM did not alter the expression of GFP or gD transcripts in TG of latently-infected mice.

### Effect of blocking gD-HVEM binding on T cell exhaustion in latently-infected mice

We previously reported that LAT is associated with increased reactivation and, consequently, greater CD8 T cell exhaustion [39]. Our results (Fig 3) suggest that LAT expression in TG from v27- and v27-29- infected mice is associated with

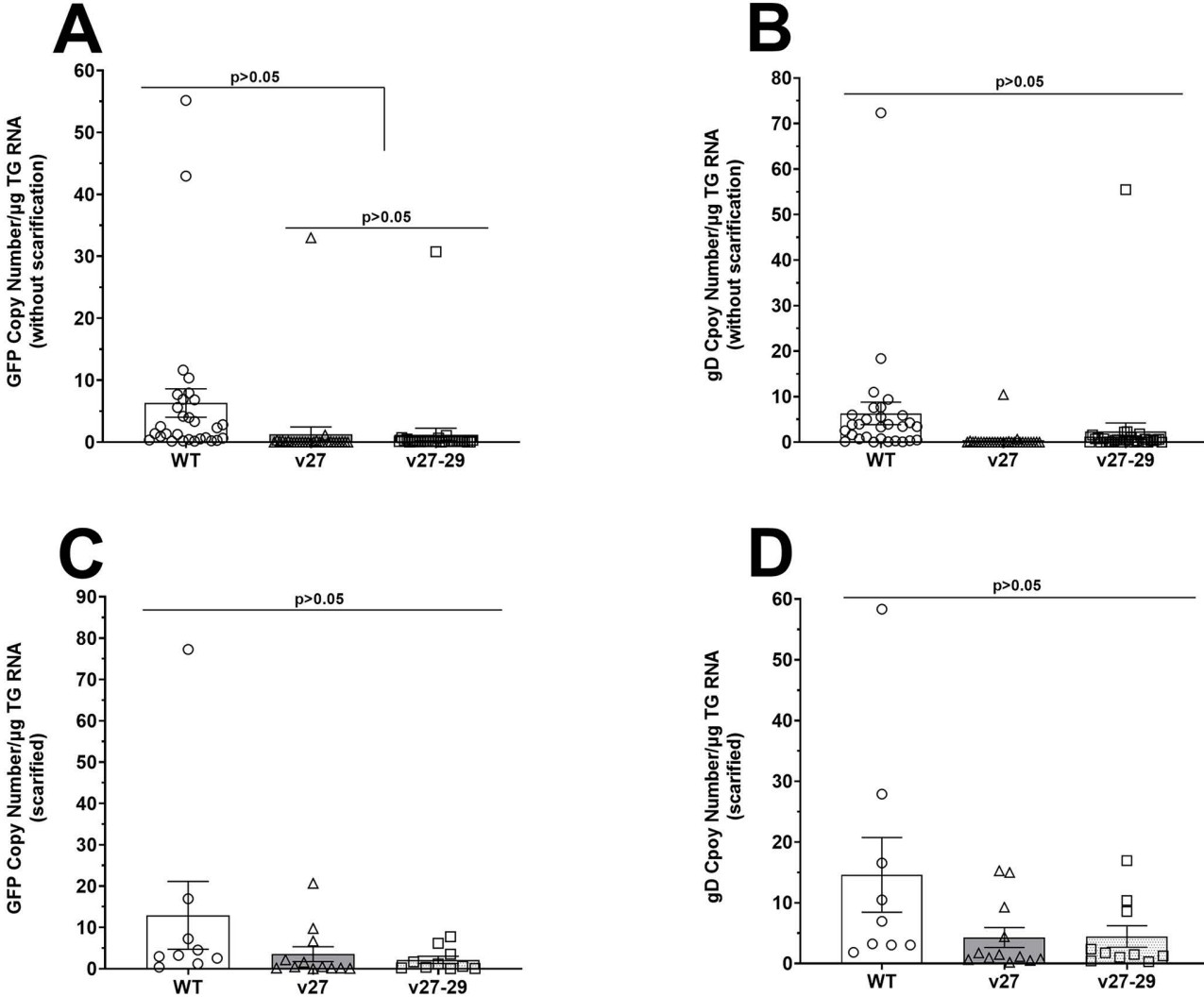

**Fig 4. Loss of gD binding to HVEM affects GFP and gD expression in TG of latently-infected mice.** RNA isolated from TG of latently-infected mice (see Fig 3), with and without corneal scarification, were used to measure GFP and gD expression in TG of latently-infected mice. **(A-B)** GFP and gD expression in latently-infected TG without scarification. qRT-PCR was performed using total RNA from individual TG (see Fig 3). Estimated copy numbers of GFP and gD were calculated from standard curves generated with pUC57-GFP-gD and pGem-gD1, respectively. GFP and gD expression were normalized to endogenous control GAPDH and plotted as mean±SEM. Data are from two independent experiments using 29, 28, and 30 TG from WT-, v27-, and v27-29-infected mice, respectively. **(C-D)** GFP and gD expression in latently-infected TG with scarification. RNA isolated from the individual TG described in Fig 3 was used for qRT-PCR to measure GFP and gD expression in TG of mice with scarification. GFP and gD copy numbers were measured as above and normalized to GAPDH. Data are mean±SEM from 9, 12, and 10 TG from WT-, v27-, and v27-29-infected mice, respectively.

reduced gD binding to HVEM. Therefore, we asked how the absence of gD binding to HVEM affects CD8 T cell exhaustion. qRT-PCR of RNA isolated from WT, v27, and v27-29 infected mice (see Fig 3) was used to measure CD8α and PD-1 transcript levels in TG of latently-infected mice. Results are presented as "fold" increase over baseline RNA levels in TG of naive mice (Fig 5). In TG of mice latently-infected with v27- and v27-29 viruses without scarification, CD8α RNA levels were significantly less than in WT-infected mice (Fig 5A; $p < 0.0001$, Fisher's exact test), while no differences in CD8α RNA levels were noted between v27 and v27-29 infected mice (Fig 5A, $p > 0.05$). Similar to CD8α RNA expression

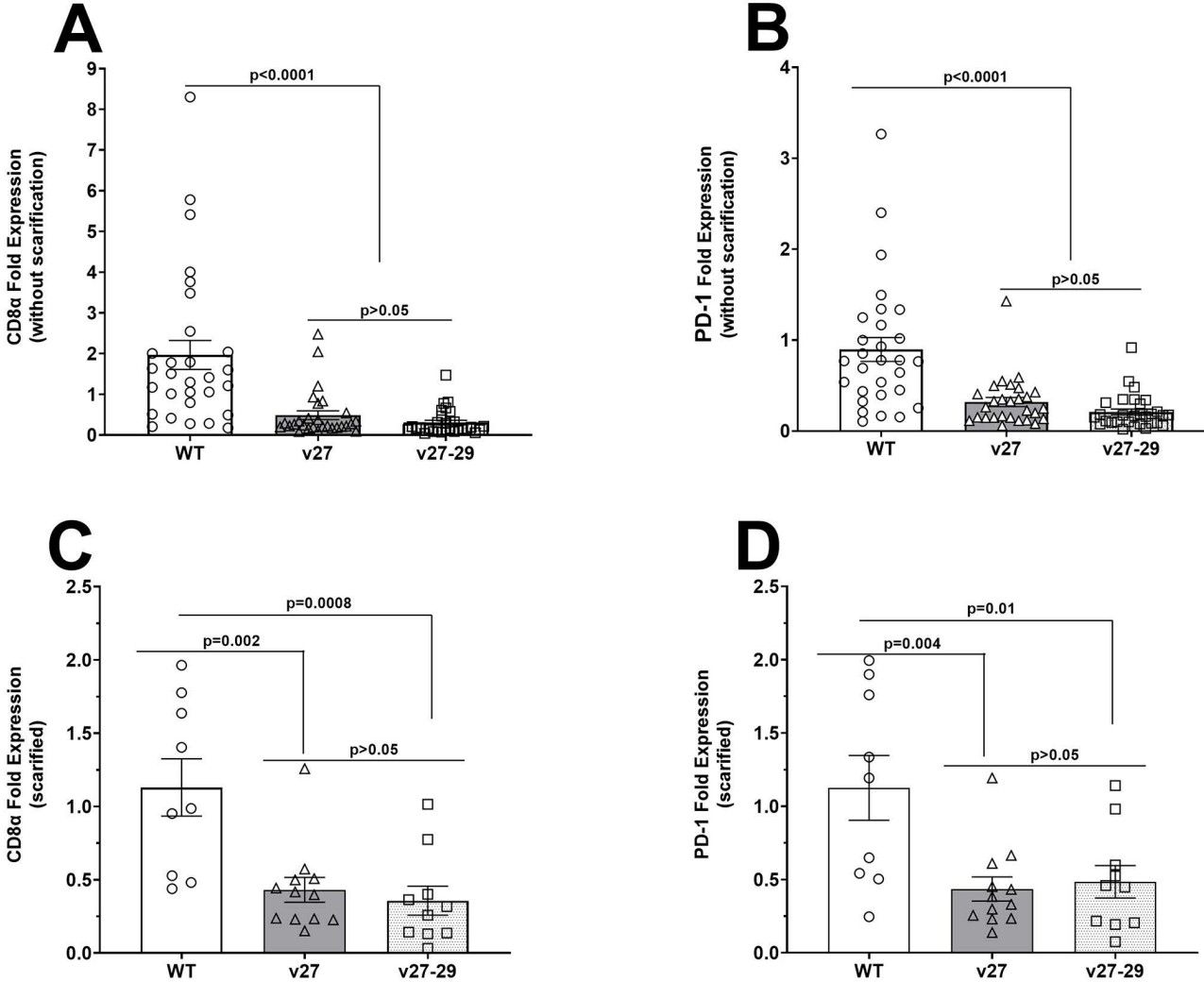

**Fig 5. Loss of gD binding to HVEM affects T cell exhaustion in TG of latently-infected mice.** RNA isolated from TG of latently-infected mice (see Fig 3), with and without corneal scarification, was used to measure CD8α and PD-1 expression in TG of latently-infected mice. **(A-B)** CD8α and PD-1 expression in latently-infected TG without scarification. qRT-PCR was performed on individual TG using total RNA. CD8α and PD-1 expression in TG from naive mice served as baseline controls. GAPDH expression was used to normalize the relative expression of each transcript in TG of latently-infected mice. Data are mean ± SEM from two independent experiments using 29, 28, and 30 TG from WT-, v27-, and v27-29-infected mice, respectively. **(C-D)** CD8α and PD-1 expression following corneal scarification. qRT-PCR was performed on individual TG as above, using CD8α and PD-1 expression in TG from naive mice to estimate relative expression of CD8α and PD-1 transcripts in TG of infected mice. GAPDH expression was used to normalize transcript levels. Data are mean ± SEM from 9, 12, and 10 TG from WT-, v27-, and v27-29-infected mice, respectively.

(Fig 5A), PD-1 RNA expression was also significantly lower in TG of v27 and v27-29 infected virus than in WT-infected mice (Fig 5B; p < 0.0001, Fisher's exact test).

We next examined expression of CD8α and PD-1 RNA in TG of infected mice with corneal scarification (Fig 5C and 5D). Similar to non-scarified mice, both CD8α (Fig 5C) and PD-1 (Fig 5D) RNA levels in v27 and v27-29 infected mice were significantly less than in WT-infected mice (Fig 5C and 5D, p < 0.0001). No differences in CD8α or PD-1 expression were seen in v27 and v27-29 infected mice (Fig 5C and 5D, p > 0.05). Consistent with our previous findings, these results

suggest that gD binding to HVEM positively correlates with higher viral replication in the eye, latency, and consequently, exhaustion.

**Blocking the binding of gD to HVEM affects reactivation in latently-infected mice**

We previously reported that HVEM is required for WT levels of latency-reactivation [23]. Thus, to determine whether blocking gD binding to HVEM will affect reactivation, mice were infected with 2 X 10⁵ pfu/eye of v27, v27-29, and WT viruses without scarification. TG were harvested from infected mice on day 28 PI, and the kinetics of virus reactivation were measured in explanted TG. The average reactivation time from WT-infected mice was 4.4±0.3 days, and from v27 and v27-29 viruses it was 6.0±0.8 and 5.1±0.5 days, respectively, and these differences were not significant (Fig 6A, p>0.05). The absence of significant differences between the three groups was associated with the number of TG reactivated in each group (Fig 6B). In WT-infected mice, 2 of 29 TG did not reactivate by day 12 post TG culture, in v27-infected mice, 26 of 30 TG did not reactivate, and in v27-29 infected mice,15 of 30 TG did not reactivate. Differences in the number of TG that did not reactivate among the three groups were statistically significant (Fig 6B). Thus, the absence of significant numbers of TG that did not reactivate suggests that binding of gD to HVEM is required for efficient reactivation and depends on virus load in the eye of infected mice.

We next asked whether corneal scarification increased the reactivation rate in v27- and v27-29-infected mice when compared with WT virus-infected mice. After corneal scarification, mice were ocularly infected with WT, v27- or v27-29 viruses as described above. The average reactivation time from mice infected with WT virus was 3.8±0.1 days, 4.6±0.3 days for v27-infected mice, and 4.6±0.1 days for v27-29-infected mice (Fig 6C). The reactivation rate was statistically significant between WT-infected and v27-infected mice (Fig 6C, p=0.009) as well as between WT-infected and v27-29 infected mice (Fig 6C, p=0.01), while no significant differences were detected between v27 and v27-29 infected mice (Fig 6C, p>0.05). These results suggest that corneal scarification enhances reactivation in v27- and v27-29-infected mice, indicating that binding of gD to HVEM is required for WT-level reactivation.

## Discussion

We previously reported that *HVEM*⁻/⁻ mice exhibit significantly reduced HSV-1 latency and reactivation compared with WT mice, whereas primary infection is similar in both strains, suggesting that HVEM plays an important role in latency and reactivation in ocularly infected mice [23]. In published studies, aa Q27 of gD was shown to disrupt gD binding to HVEM, but not to Nectin-1 [40]. Although mutant viruses, including KOS-Rid1, KOS-Rid2, and HSV-1 strain ANG, have been used to study gD binding to HVEM [25–27,41], in contrast to McKrae, these viruses require corneal scarification due to their lower infectivity *in vivo*. Thus, we asked whether, as in HSV-1 strain KOS [40], mutation of gD aa 27 in HSV-1 strain McKrae would completely block gD binding to HVEM. In contrast to published reports, we found that complete blocking of gD to HVEM required mutations in aa 27–29 of gD [30]. Because this region of gD is highly conserved in HSV-1 various strains (*i.e.*, KOS, F, McKrae, 17, H129, SC, ANG) [42–45], we constructed two mutant viruses: one with a single Q27P mutation (v27) and the other with a three aa mutation (Q27A-L28A-T29A) in gD (v27-29) [30]. Similar to our previous study with KOS-Rid1 and KOS-Rid2 [25], ocular infection of mice with the v27 and v27-29 viruses, with or without corneal scarification, produced significantly less ocular viral replication than with WT virus. Although corneal scarification resulted in more virus replication in the eyes of v27 and v27-29 infected mice than in mice infected ocularly without scarification, virus replication was still lower than in mice infected with WT virus. These differences occurred even though replication of the v27 and v27-29 viruses was similar to that of the WT virus across four cell lines *in vitro* [30]. In contrast to lower replication of v27 and v27-29 viruses in the eyes of C57BL/6 mice, McKrae virus replication in the eyes of these mice was similar to that in *HVEM*-/- and naive C57BL/6 infected mice [23]. Thus, using viral replication in the eyes of infected mice as a readout, blocking the interaction between gD and HVEM provides a means to distinguish the effects of intact viral gD from those of host HVEM deficiency in the same infection model. Additionally, the gD Q27 mutation does not affect

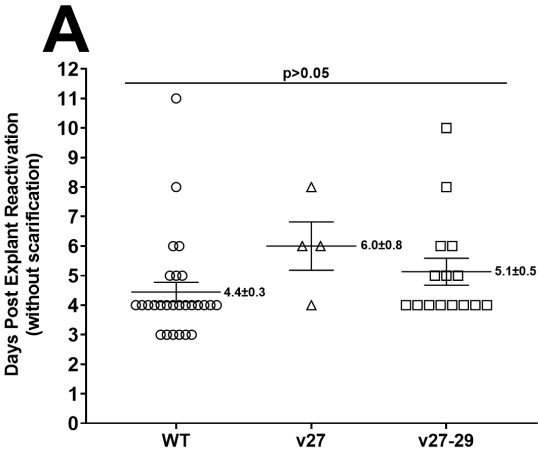

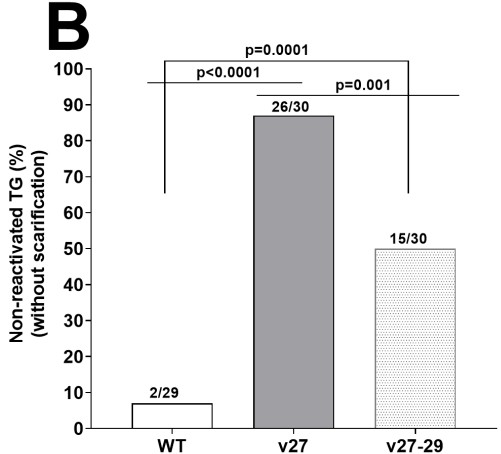

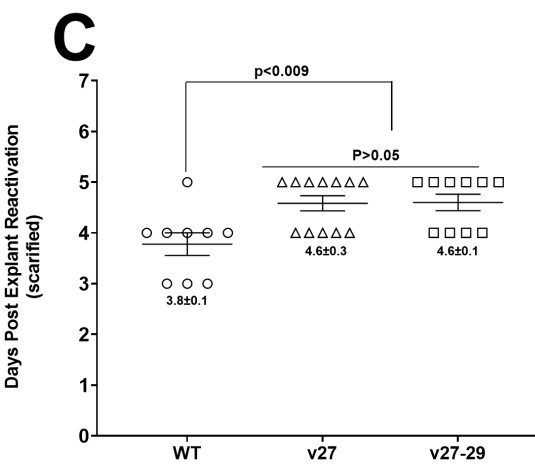

**Fig 6. Loss of gD binding to HVEM affects reactivation in TG of latently-infected mice. (A)** Explant reactivation in TG of latently-infected mice without scarification. Mice were ocularly infected with $2 \times 10^5$ pfu/eye of WT, v27, or v27-29 virus without corneal scarification. Individual TG was harvested on day 28 PI and incubated individually in 1.5 mL of tissue culture medium at 37°C. Media aliquots were collected daily for up to 5 days from each culture and plated onto RS indicator cells to assess viral reactivation. Results were plotted as the number of TG that reactivated each day. Numbers indicate the mean time of reactivation ± SEM for each group. Data are from two independent experiments using 27, 4, and 15 TG from WT-, v27-, and v27-29-infected mice, respectively. P-values were determined using one-way ANOVA. **(B)** Percentage of TG from (A) that did not reactivate in explant

cultures. Data are from two independent experiments based on the number of TG that did not reactivate in explant cultures: 2 of 29 (7%), 26 of 30 (87%), and 15 of 30 (50%) TG did not reactivate from WT-, v27-, and v27-29-infected mice, respectively. **(C)** Reactivation following corneal scarification in infected mice. Mice were ocularly infected with 2 X $10^5$ pfu/eye of WT, v27, or v27-29 virus with corneal scarification, and individual TG was harvested on day 28 PI. Explant reactivation was performed on isolated TG, and the time of reactivation was recorded as above. Data are mean ± SEM from 9, 12, and 10 TG from WT-, v27-, and v27-29-infected mice, respectively. All TG from scarified mice were reactivated.

gD binding to Nectin-1 [27,46]. Given the diversity of HSV-1 entry routes *in vitro* and *in vivo*, the absence of HVEM in the acute phase of infection is less critical than in the latent phase [23]. Conversely, the absence of gD binding to HVEM is important in both primary and latent infection. Thus, effects attributed to the absence of gD-HVEM binding during latency may instead result from the absence of HVEM itself, because HVEM can modulate T cell activation via its interactions with BTLA and LIGHT [24]. This dual role implies that gD-HVEM interactions influence viral infectivity and immune responses that shape both primary and latent infection.

In this study, we show that partial or complete blocking of gD binding to HVEM, with or without scarification, produced significantly less CS and angiogenesis in infected mice than in mice infected with the WT virus. Although scarification increased viral replication in the eyes of v27- and v27-29-infected mice over that in unscarred controls, it did not increase CS or angiogenesis. However, our corneal scarification experiments did not establish a link between receptor utilization and disease severity. Thus, gD binding to HVEM increases eye disease due to higher virus replication resulting from gD binding to HVEM in infected mice. This is consistent with our previous reports showing that virus load directly correlates with the duration of primary virus replication in the eye and the severity of eye disease in ocularly infected WT mice [8,9,47,48]. In this study, we further found that WT gD, acting directly via HVEM binding, contributed to enhancing eye disease in infected mice. Mutations at gD aa 27 or aa 27–29 attenuated v27 and v27-29 viruses, respectively, suggesting that weakening the gD-HVEM interaction limits viral replication while also blunting inflammatory cascades responsible for ocular disease, likely by altering HVEM-mediated immune modulation at the site of infection.

In this study, reduced viral replication during primary infection produced a profound difference in latency establishment, as measured by LAT expression levels. LAT expression was significantly lower in TG from mice infected with v27 or v27-29 than in WT-infected mice. Corneal scarification in infected mice increased LAT expression in the three groups of infected mice by approximately twofold over their unscarified counterparts. Reduced LAT levels were similar in scarified and non-scarified conditions, suggesting that scarification enhances local infection, but does not rescue the deficiency in gD-HVEM signaling relative to WT HSV-1. Previously, we reported that LAT levels in *HVEM*-/- mice were significantly lower than in WT mice [23]. Our current results also indicate that gD-HVEM signaling plays a significant role in HSV-1 latency. Similar to *HVEM*-/- mice [23], TG from mutant virus-infected mice expressed lower CD8α and PD-1 markers that are linked to T-cell exhaustion and chronic stimulation during latency [39]. Previously, we reported that higher latency correlated with higher levels of CD8 and PD-1 RNA in TG of latently-infected mice [49], indicating that higher levels of HSV-1 latency are associated with increased CD8 T cell exhaustion. The absence of gD binding to HVEM may also alter HVEM interactions with BTLA and LIGHT, thereby reshaping T cell exhaustion. However, we previously reported that, unlike HVEM expression, HSV-1 infection did not significantly alter LIGHT or BTLA RNA levels [23]. This is consistent with the idea that LIGHT and BTLA expression occurs in immune cells within the microenvironment of the latently-infected cell and thus, is not affected by LAT expression in latently-infected neurons [24].

In contrast to primary infection and consistent with latency levels, we have shown that HSV-1 reactivation is significantly lower in *HVEM*-/- mice, suggesting that HVEM is involved in latent virus reactivation [23]. As in *HVEM*-/- mice [23], *ex vivo* reactivation assays in this study support the attenuated phenotype of gD mutant viruses. Without corneal scarification, the average time to reactivation was similar among groups, although the number of reactivated TG was significantly lower in v27- and v27-29-infected mice than in mice infected with WT virus. In corneal-scarified animals, all v27 and v27-29 infected TG reactivated, and the time to reactivation was increased in v27 and v27-29-infected mice than in

WT-infected mice. Consistent with this study, using KOS-Rid1 and KOS-Rid2 viruses, we reported that HSV-1 reactivation from latency is significantly delayed in the absence of gD binding to HVEM [25]. However, in contrast to viruses in the KOS background, our results in McKrae backgrounds showed that the absence of gD binding to HVEM affects both primary infection and latency-reactivation in infected mice. These results illustrate the importance of using the proper virus background to study the gD-HVEM relationship. Our results are consistent with prior work showing that disruption of the gD-HVEM interaction compromises neuroinvasion and reactivation efficiency [50]. The triple v27-29 mutant consistently had a stronger reduction phenotype than the single-substitution v27 virus, confirming that aa Q27-T29 cooperatively sustain productive receptor engagement and downstream signaling events needed for efficient replication and subsequent latency-reactivation. Thus, these aa are critical determinants of HSV-1 virulence that operate beyond entry to influence immune pathogenesis and latency dynamics [25].

Taken together, our findings delineate the integral role of HVEM engagement in driving virulence, ocular disease severity, latency, and reactivation. The contrasting outcomes in scarified and non-scarified conditions demonstrate that mechanical facilitation of viral entry does not fully replace receptor-mediated signaling during infection. Efficient gD-HVEM binding likely enhances early virus-host interactions that coordinate epithelial spread, neuroinvasion, and the inflammatory milieu conducive to viral persistence [24,51]. The reduced corneal pathology associated with mutant infections suggests that impaired HVEM engagement dampens local cytokine release and leukocyte recruitment, thereby blunting the inflammatory response and attenuating ocular damage. While Nectin-1 and 3-OS-HS may partially compensate for entry in scarified mice, HVEM appears to play a unique role in promoting efficient epithelial infection and immune modulation. As seen with the diminished ability of mutants to replicate and reactivate, receptor-specific engagement not only determines viral tropism but also governs downstream effects on latency control [25,50]. These findings position the gD-HVEM interface as a central determinant of HSV-1 pathogenesis and highlight this axis as an attractive target for therapeutic or vaccine strategies to limit entry.

## Materials and methods

### Ethics statements

All procedures were performed in strict accordance with the Association for Research in Vision and Ophthalmology Statement for the Use of Animals in Ophthalmic and Vision Research and the NIH Guide for the Care and Use of Laboratory Animals (ISBN 0-309-05377-3). The animal research protocol was approved by the Institutional Animal Care and Use Committee of Cedars-Sinai Medical Center (protocol no. 8837).

### Cells, viruses, and mice

Rabbit skin (RS) cells were used to prepare virus stocks, culture mouse tear swabs, and determine viral growth kinetics. RS cells were grown in Eagle's minimal essential medium supplemented with 5% fetal bovine serum. Triple plaque purified GFP-McKrae, v27, and v27-29 mutant viruses were used in this study. GFP-McKrae, v27, and v27-29 viruses were described previously [30,38]. Previously, we constructed a recombinant HSV-1 virus expressing GFP upstream of gD in a WT-McKrae background, referred to as GFP-McKrae [38]. Based on WT-McKrae nucleotides 136441–139321, we synthesized two plasmids: one with a single aa mutation in gD aa 27 (Q27P); and the second with three aa mutations (Q27A-L28A-T29A). Both plasmids contain a complete GFP sequence under the control of a cytomegalovirus (CMV) promoter, flanked by two unique Pac1 sites (GenScript, Piscataway, NJ). These constructs were inserted into the pUC57 Pac1 site, and the resulting plasmids were designated pUC57–1Xmut-gD (Q27P) and pUC57–3Xmut-gD (Q27A-L28A-T29A). In all the constructs, CMV-driven EGFP was placed upstream of gD with the desired mutations. Recombinant viruses v27 (q27P Q27P mutation) and v27-29 (Q27A-L28A-T29A) were generated by co-transfecting pUC57–1Xmut-gD or pUC57–3Xmut-gD together with infectious McKrae DNA using the calcium phosphate method as described [30,38]. Viruses from the co-transfection were plated, and isolated plaques were picked and screened for GFP expression as we

described [30,38]. Selected plaques with GFP gene expression were plaque-purified five times. Presence of the expected gD mutations was confirmed by PCR and next-generation sequencing. Complete genome sequences of v27 (GenBank accession- PX763612) and v27-29 (GenBank accession PX837190) have been deposited in GenBank. Six to 7-week-old female C57BL/6 mice were used (Jackson Laboratory, Bar Harbor, ME).

## Ocular infection

Mice were infected via the ocular route with 2 X $10^5$ pfu/eye of each virus suspended in 5 μl of tissue culture medium (supplemented with 5% serum). Viruses were administered as eye drops with and without corneal scarification.

## Virus titration in tears of infected mice

Tear films were collected from both eyes of each ocularly infected mouse on days 1–5 PI using a Dacron-tipped swab. Each swab was placed in 0.5 ml of tissue culture medium, squeezed, and the virus titer was determined using a standard plaque assay on RS cells.

## HSV-induced eye disease and angiogenesis

The severity of corneal scarring and angiogenesis in mouse corneas was examined by slit lamp biomicroscope on day 28 PI. The examination was conducted by an investigator blinded to the mouse treatment regimens. Mice from each group were randomly screened for haziness without anesthesia. The scale was: 0, normal cornea; 1, mild haze; 2, moderate opacity; 3, severe corneal opacity but iris visible; 4, opaque and corneal ulcer; and 5, corneal rupture and necrotizing keratitis, as we previously described [25]. The severity of angiogenesis on day 28 PI was recorded using a 4 pt scale in which a grade of 4 for a given quadrant of the circle represents centripetal growth of 1.5 mm toward the corneal center. Scores from the four eye quadrants were summed to derive the neovessel index (range, 0–16) for each eye at a given time point. Data was then plotted in Prism to generate a graph.

## TG RNA extraction, cDNA synthesis, and TaqMan RT-PCR

TG from latently infected mice were collected on day 28 PI, immersed in RNAlater RNA stabilization reagent, and stored at -80°C for processing. Qiazol RNA reagent (Qiagen) and 1-Bromo-3-chloropropane (BCP Cat. No.-B9673, Sigma-Aldrich) were used to extract RNA from individual TG. Total RNA was extracted as we described previously [52]. Following RNA extraction, 1000 ng of total RNA was reverse-transcribed using random hexamer primers and murine leukemia virus reverse transcriptase from the High-Capacity cDNA Reverse Transcription Kit (Applied Biosystems, Foster City, CA), according to the manufacturer's recommendations. Primer probe sets consisted of two unlabeled PCR primers and the FAM dye-labeled TaqMan MGB probe formulated into a single mixture. The following two assays were used in this study: 1) CD8α (ABI Mm01182108_m1, amplicon length = 67 bp); and 2) PD-1 (programmed death 1; ABI Mm00435532_m1; amplicon length = 65 bp). In all experiments, glyceraldehyde-3-phosphate dehydrogenase (GAPDH, Mm99999915_g1, Amplicon size 107 bp) was used to normalize transcripts. For GFP, a custom TaqMan Gene Expression Assay (Catalog number: 4331348; Assay ID: AP9HURV) was used, as described previously [38]. The $2^{-\Delta\Delta CT}$ method was used to calculate fold change in gene expression relative to expression in WT-infected controls.

Levels of gD and LAT RNA in latent TG were determined using two custom-made primer and probe sets. 1) gD Forward: 5'-GCGGCTCGTGAAGATAAACG-3', gD Reverse: 5'-CTCGGTGCTCCAGGATAAACTG-3', gD Probe: 5'-FAM-CTGGACGGAGATTACA-3', and gD Amplicon length = 59 bp. 2) LAT Forward- 5'-GGGTGGGCTCGTGTTA'AG-3,' LAT Reverse- 5'-GGACGGGTAAGTAACAGAGTCT'TA-3'; LAT Probe- 5'-FAM-ACACCAGCCCGTTC'TT-3,' and LAT Amplicon length = 81 bp. Relative copy numbers for gD and LAT were calculated using standard curves generated from plasmids pAc-gD1 and pGEM5317, respectively [38,53].

### *In vitro* explant reactivation assay

TG from latently infected mice were removed on day 28 PI and cultured in tissue culture medium as previously described [38]. Briefly, a 100-µl aliquot was taken daily from each culture and used to infect RS cell monolayers. Infected RS cells were monitored daily for 5 days for the appearance of cytopathic effect to determine the time at which the reactivated virus from each TG first appeared. Because media from explanted TG cultures were plated daily, the day on which the reactivated virus first appeared in the explanted TG cultures could be determined.

### Statistical analysis

For all statistical tests, *p*-values less than or equal to 0.05 were considered statistically significant and are indicated by a single asterisk (*), and *p*-values less than or equal to 0.001 are indicated by double asterisks (**). A two-tailed Student's t-test with unequal variances was used to compare the two experimental groups. A one-way analysis of variance (ANOVA) was used to compare differences among three or more experimental groups. All experiments were repeated at least three times to ensure accuracy.

### Author contributions

**Conceptualization:** Deepak Arya, Homayon Ghiasi.

**Data curation:** Deepak Arya, Ujjaldeep Jaggi, Jay J Oh, Shaohui Wang.

**Formal analysis:** Deepak Arya, Ujjaldeep Jaggi, Homayon Ghiasi.

**Funding acquisition:** Homayon Ghiasi.

**Investigation:** Homayon Ghiasi.

**Methodology:** Deepak Arya, Ujjaldeep Jaggi, Shaohui Wang.

**Project administration:** Homayon Ghiasi.

**Resources:** Shaohui Wang, Homayon Ghiasi.

**Software:** Deepak Arya.

**Supervision:** Homayon Ghiasi.

**Validation:** Deepak Arya, Homayon Ghiasi.

**Visualization:** Deepak Arya, Ujjaldeep Jaggi, Homayon Ghiasi.

**Writing – original draft:** Deepak Arya, Homayon Ghiasi.

**Writing – review & editing:** Deepak Arya, Homayon Ghiasi.

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
