## [Decision Letter · Decision Letter 0]

7 Apr 2026

PPATHOGENS-D-26-00687

Partial or complete blocking of gD binding to HVEM affects primary and latent HSV-1infection as well as eye disease in HSV-1 infected mice

PLOS Pathogens

Dear Dr. Ghiasi,

Thank you for submitting your manuscript to PLOS Pathogens. After careful consideration, we feel that it has merit but does not fully meet PLOS Pathogens's publication criteria as it currently stands. Therefore, we invite you to submit a revised version of the manuscript that addresses the points raised during the review process.

We look forward to receiving your revised manuscript.

Kind regards,

Deepak Shukla

Academic Editor

PLOS Pathogens

Blossom Damania

Section Editor

PLOS Pathogens

Sumita Bhaduri-McIntosh

Editor-in-Chief

PLOS Pathogens

orcid.org/0000-0003-2946-9497

Michael Malim

Editor-in-Chief

PLOS Pathogens

orcid.org/0000-0002-7699-2064

**Journal Requirements:**

1) We do not publish any copyright or trademark symbols that usually accompany proprietary names, eg ©,  ®, or TM  (e.g. next to drug or reagent names). Therefore please remove all instances of trademark/copyright symbols throughout the text, including:

- TM on page: 16.

2) We note that your Data Availability Statement is currently as follows: "All relevant data are in the manuscript.". Please confirm at this time whether or not your submission contains all raw data required to replicate the results of your study. Authors must share the “minimal data set” for their submission. PLOS defines the minimal data set to consist of the data required to replicate all study findings reported in the article, as well as related metadata and methods (https://journals.plos.org/plosone/s/data-availability#loc-minimal-data-set-definition).

3) Please ensure that the funders and grant numbers match between the Financial Disclosure field and the Funding Information tab in your submission form. Note that the funders must be provided in the same order in both places as well. Currently, the order of the grants is different in both places.

**Reviewers' Comments:**

Reviewer's Responses to Questions

**Part I - Summary**

Reviewer #1: Herpes simplex virus type 1 (HSV-1) glycoprotein D (gD) plays a major player of virus entry, in large part because it interacts with heparan, HVEM. Previous studies revealed that amino acid 27-29 completely blocked gD binding to HVEM. This was the rational for mutating amino acid 27 and a mutant virus aa 27-29. These mutants were in the McKrae strain of HSV-1, which is a highly virulent strain of HSV-1. These mutant viruses were used to compare viral replication, corneal disease, latency, reactivation form latency, and T cell exhaustion. This study revealed that the ability of gD to interact with HVEM is crucial for viral replication and reactivation from latency. In general, this is an interesting study because interfering gD-HVEM can be used to develop a vaccine or small molecules that blocks the binding of HVEM to gD. With that said, there are a few minor issues that need to be addressed.

Reviewer #2: Summary: Using a well characterized mouse model of HSV1 corneal infection, latency, and reactivation, Arya and coworkers performed a series of studies to investigate with greater precision the binding of virus-encoded glycoprotein D (gD) to host cell receptor HVEM based on an observation that gD amino acids Q27 – T29 play a central role in blocking gD binding to HVEM. Thus, two mutant viruses on a McKrae virus strain background were constructed, one with a single gD amino acid mutation (v27) and the other with three gD mutations (v27 – 29). The v27 mutant virus, the v27 – 29 mutant virus, and WT McKrae virus constructed to express GFP (GFP-McKrae) were compared for a number of experimental endpoints that included amounts of infectious virus in tear films, corneal scarring, angiogenesis, copies of LAT per ug of trigeminal ganglion RNA, copies of GFP and gD per ug of trigeminal ganglion RNA, CD8alpha and PD-1 expression, and reactivation of virus from explanted trigeminal ganglia. Importantly, these experimental endpoints were determined in mice with or without corneal scarification prior to HSV1 infection. Overall, v27 and v27 – 29-infected mice showed reduced amounts for all experimental endpoints examined when compared with mice infected with WT GFP-McKrae virus. Importantly, corneal scarification did indeed impact findings with mice with scarified corneas showing increased amounts for most experimental endpoints examined when compared with mice not subjected to corneal scarification. The authors conclude that disrupting gD-HVEM reduces virus virulence and limits latency, reactivation, and T cell exhaustion. The Q27 – T29 region of gD is therefore a critical determinant of infection.

Review: This is a well written manuscript that summarizes the findings of a series of relatively straightforward studies that provide convincing outcomes. These studies represent a nice extension of previous work by this research group that provides new and important information on how gD-HVEM interaction affects acute HSV1 replication at the corneal surface (especially if scarified or not scarified) as well as HSV1 latency and reactivation. Attention to a few issues might improve the manuscript and make it more reader friendly, however.

1. The authors surprisingly use GFP expression as an experimental endpoint to compare v27 v27 – 29, and WT McKrae viruses (Fig 4). More information is therefore needed about the precise construction of WT GFP-McKrae in the Materials and Methods section even though a previous paper from the literature is cited. This is needed because the GFP molecule is not naturally encoded by HSV1 and this molecule expression could be considered highly artificial. Where in the virus genome is the GFP gene placed? Is it expressed as an immediate-early, early, or late gene product? Would use of a virus-encoded immediate early or early gene product serve better than GFP expression from an artificially recombinant virus genome?

2. It is noteworthy that a number of figures show outlier values (see Fig 4 and 5). Would the authors please comment on these outlier values and the possible reason(s) for their appearance?

3. Line 37 of the Abstract states “… that disrupting gD-HVEM inhibition reduces viral virulence …” Is use of the word “inhibition” correct in this sentence?

Reviewer #3: In this study, the authors used HSV-1 McKrae strain with mutations in gD protein, which prevented gD binding efficacy to HVEM. As a result, the authors showed reduced viral virulence, latency, reactivation and T cell exhaustion. The study used mouse model of corneal HSV-1 infection with or without corneal scratching. This is an interesting study, which highlights the outcome of gD-HVEM interaction during corneal HSV-1 infection. There are few minor concerns that authors should address to enhance the impact of their findings-

1. The authors should provide HSV-1 strain name in Figure 1 legend.

2. What was the rationale of choosing D28 post-infection for the clinical scoring of disease?

3. HSV-1 McKrae is a highly neurovirulent strain and after corneal scratching, it may induce encephalitis. Did the authors note encephalitis in the corneal scratching group?

4. What could be the possible reason of reduced PD-1 level on T cells in the TG of mice receiving mutant virus?

**Part II – Major Issues: Key Experiments Required for Acceptance**

Reviewer #1: No major issues.

Reviewer #2: None

Reviewer #3: None

**Part III – Minor Issues: Editorial and Data Presentation Modifications**

Reviewer #1: (No Response)

Reviewer #2: None

Reviewer #3: Described in Part I

PLOS authors have the option to publish the peer review history of their article (what does this mean?). If published, this will include your full peer review and any attached files.

Reviewer #1: No

Reviewer #2: No

Reviewer #3: No

**Figure resubmission:**
---

## [Decision Letter · Decision Letter 1]

13 May 2026

Dear Dr. Ghiasi,

We are pleased to inform you that your manuscript 'Partial or complete blocking of gD binding to HVEM affects primary and latent HSV-1 infection as well as eye disease in HSV-1 infected mice' has been provisionally accepted for publication in PLOS Pathogens.

Best regards,

Deepak Shukla

Academic Editor

PLOS Pathogens

Blossom Damania

Section Editor

PLOS Pathogens

Sumita Bhaduri-McIntosh

Editor-in-Chief

PLOS Pathogens

orcid.org/0000-0003-2946-9497

Michael Malim

Editor-in-Chief

PLOS Pathogens

orcid.org/0000-0002-7699-2064

Reviewer Comments (if any, and for reference):

Reviewer's Responses to Questions

**Part I - Summary**

Reviewer #1: For this revised manuscript, the author revealed that the HSV-1 glycoprotein D (gD) plays a major player of virus entry, in large part because it interacts with heparan, HVEM. Previous studies revealed that amino acid 27-29 completely blocked gD binding to HVEM. This was the rational for mutating amino acid 27 and a mutant virus aa 27-29. These mutants were in the McKrae strain of HSV-1, which is a highly virulent strain of HSV-1. These mutant viruses were used to compare viral replication, corneal disease, latency, reactivation form latency, and T cell exhaustion. This study revealed that the ability of gD to interact with HVEM is crucial for viral replication and reactivation from latency.

Reviewer #2: The authors have responded thoroughly to all concerns and questions.

Reviewer #3: The authors have addressed the concerns raised in the original submission. The revised version of the manuscript is now suitable for publication in this journal.

**Part II – Major Issues: Key Experiments Required for Acceptance**

Reviewer #1: None

Reviewer #2: Not applicable

Reviewer #3: Not required.

**Part III – Minor Issues: Editorial and Data Presentation Modifications**

Reviewer #1: The authors addressed the minor concerns that I raised.

Reviewer #2: The authors have responded thoroughly to all concerns and questions.

Reviewer #3: None

PLOS authors have the option to publish the peer review history of their article (what does this mean?). If published, this will include your full peer review and any attached files.

Reviewer #1: No

Reviewer #2: No

Reviewer #3: **Yes:**Susmit Suvas

---

## [Editor Report · Acceptance letter]

Dear Dr. Ghiasi,

We are delighted to inform you that your manuscript, "Partial or complete blocking of gD binding to HVEM affects primary and latent HSV-1 infection as well as eye disease in HSV-1 infected mice," has been formally accepted for publication in PLOS Pathogens.

Best regards,

Sumita Bhaduri-McIntosh

Editor-in-Chief

PLOS Pathogens

orcid.org/0000-0003-2946-9497

Michael Malim

Editor-in-Chief

PLOS Pathogens

orcid.org/0000-0002-7699-2064